# Bereavement practices and staff competencies on perinatal loss at a community-based teaching hospital

Carolina Garavito[1], Shweta Karki[2], Suzanne J. Rose[2]*, Josette Hartnett[3]

**1** Bistrol Myers Squibb, Clinical Research Associate, Regionally based in Connecticut, Bridgeport, Connecticut, United States of America, **2** Department of Research and Discovery, Stamford Hospital, Stamford, Connecticut, United States of America, **3** Burke Rehabilitation Hospital, White Plains, New York, United States of America

\* srose@stamhealth.org

## Abstract

### Objective

Although considerable research exists on how families experience perinatal loss, the bereavement competencies and practices at our hospital were unknown. To investigate the baseline bereavement knowledge, skills, self-awareness, and organizational support among staff working at a community-based teaching hospital, a cross-sectional study was conducted using the validated Perinatal Bereavement Care Confidence Scale (PBCCS).

### Methods

PBCCS scores range from 1 to 5 (1 representing Strongly Disagree and 5 representing Strongly Agree). Bereavement support confidence was calculated by using mean score for each subscale to obtain total PBCCS score (score range of 32–160). A sufficient level of confidence is considered 80% or higher. Independent t-tests/ANOVA measured differences in scores for two or more groups.

### Results

A total of 46 survey responses were received; 19 were submitted from Labor and Delivery, 13 from the Emergency Department and 5 from Maternity. 28 respondents reported that they had not received any formal education on perinatal bereavement (p = 0.006). Scores for each subscale showed low levels of bereavement support knowledge (46.45 ± 13.04), skills (23.07 ± 7.31), self-awareness (24.85 ± 7.54) and organizational support (21.73 ± 6.08). The total mean sum score was 116 (72%), indicating inadequate levels of bereavement confidence overall. Those reporting having a formal education on perinatal bereavement care were more likely to have greater self-awareness as compared to those who did not (4.23 vs 3.39, p = 0.001).

**Data availability statement:** All de-identified data is publicly available within figshare at the following location: https://doi.org/10.6084/m9.figshare.30138229 and can also be provided with any written request to the corresponding author.

**Funding:** The author(s) received no specific funding for this work.

## Conclusion

Improving perinatal loss management in a hospital setting requires a comprehensive approach that includes data-driven assessment, staff education, support, and structured protocols. Key factors influencing compassionate, dignified, and holistic bereavement care provides nursing leadership with valuable information on educational gaps to target and when formalizing protocols or workflows to address this need.

## Introduction

Perinatal loss, defined as the death of a baby during pregnancy, at the time of birth, or shortly after birth, has a profound impact on expecting parents [1–3]. In the United States (U.S.), the stillbirth ratio (loss before birth at ≥20 weeks' gestation per 1,000 total births) was 5.73 per 1,000 with a total of 21,000 stillbirths reported in 2023 [4,5]. The World Health Organization (WHO) published a statement on respectful maternity care, including dignified maternal care for grieving parents [6]. Similarly, and more recently, the United Nations Interagency Group for Child Mortality Estimation (UNIGME) recommended providing the highest quality bereavement care by providing comprehensive and ongoing training and support to all members of the teams providing maternity care [7]. Despite these recommendations and guidelines, the needs of bereaved parents' are frequently unmet [8–10].

Perinatal loss has a great effect on parents and has been cited to cause intense psychological distress [1]. A recent systematic review found that the most frequently reported symptoms after stillbirth were high rates of depressive symptoms, anxiety, post-traumatic stress, suicidal ideation, panic and phobia [11]. Psychological outcomes may also depend on the healthcare professionals' ability to provide effective bereavement support, as it has been demonstrated that training and education are essential for both bereaved parents and their healthcare providers, fostering better communication, shared decision-making, and acknowledgement of parenthood [10].

Similarly, the effectiveness of the provision of bereavement care and support is vital for both long- and short-term psychological outcomes, as recently cited by qualitative focus groups and interviews with families who experienced a perinatal loss [10,12,13]. Recent research reported that healthcare workers are not specifically trained on bereavement care or processes to begin after the diagnosis is made. Therefore, psychological outcomes for parents experiencing loss may be impacted by the hospital staffs' ability to provide effective bereavement support [14–17]. Nurses in the maternity and emergency settings have reported both lack of preparation and knowledge regarding how to care for bereaved parents and/or how to address the complex circumstances surrounding perinatal loss [18]. These complexities include helping the family start a normal grieving process, actualizing the loss, being able to acknowledge their grief as part of the healing process, and meeting the particular cultural and behavioral needs of each family [19].

Strategies and programs to improve knowledge, skills, and abilities of hospital staff in bereavement care provision have been implemented in recent years. The TEARDROP Program (Teaching, Excellent, pArent, peRinatal, Deaths-related, inteRactions, tO, Professionals), a perinatal bereavement care training program based in Ireland educates practitioners on the essential skills and workflow needed for providing high-quality patient and family-centered bereavement care. Program evaluation results indicated that before the trainings, participants had low levels of bereavement skills and knowledge; post-training, care satisfaction increased for bereaved parents and providers reported feeling well equipped in caring for grieving parents [20]. Similarly, online training courses and interventions to improve bereavement support care provision were implemented in the United Kingdom with results showing notable improvements from baseline in self-reported confidence and communication skills [21].

In Connecticut, where our hospital is located, the stillbirth rate was 4.53 per 1,000 live births in 2023, equating to 168 deaths [22]. Although stillbirth affects families across all races, ethnicities, and income levels, it disproportionately impacts minority populations, with longstanding disparities seen in infant mortality rates which have grown over the last century [23]. Connecticut legislature passed a law in 2023 allowing parents to spend at least 24 hours in the hospital with their babies as they grieve their loss [24] and while validated tools to measure outcomes are reported, there is relatively little evidence based research on provider competencies surrounding perinatal loss in diverse settings.

This quality improvement needs assessment presents the historical frequencies of perinatal loss, as well as the knowledge, attitudes and processes that staff currently utilize in managing patients who experience perinatal loss at our 305-bed Magnet®- designated and Planetree certified community-based teaching hospital. 1,096 diagnoses of perinatal loss were identified at our hospital with a diagnosis code for stillbirth, miscarriage, neonatal/fetal demise, or an unplanned abortion from January 1st, 2021, to March 15th, 2024. To further understand the need for a perinatal bereavement program, this study aims to evaluate the staffs' confidence in providing perinatal bereavement support. After completing the study, the questionnaire will be sent to all hospital staff, as well as piloting interventions to address any reported gaps in knowledge, skills or confidence.

## Materials and methods

### Design, sample and setting

This cross-sectional study was conducted after receiving Institutional Review Board review and determination as exempt (WCG IRB Work Order #1-1748176-1). The survey questions can be requested from the corresponding author. The study used a convenience sample recruitment method, targeting all staff from the Maternity, Labor and Delivery (L&D) and Emergency Departments (ED). Staff working in other areas of the hospital were excluded as this project focused on the three main units that are primarily involved in the care of perinatal loss and bereaved parents/families. During the study period (March 18th, 2024-October 15th, 2024) participants were sent an email describing the study purpose, risks and benefits and the survey link, leading to the consent form and questionnaire.

### Data collection procedures

A request was submitted to the institution's data warehouse department for the number of patients with a diagnosis code for stillbirth, miscarriage, neonatal/fetal demise, or an unplanned abortion from January 1st, 2021, to March 15th, 2024. In addition, an anonymous and voluntary questionnaire was distributed via email to understand areas for improvement from the target population, including education and the need for resources. Demographic characteristics included gender, age group, highest level of education, years of working at our hospital along with information on formal education on perinatal bereavement support (yes/no). The dependent variable for this study was bereavement support confidence, measured using the PBCCS. The independent variables included current area of practice (ED, L&D, Maternity) as well as receiving formal education on perinatal bereavement support (yes/no).

### Instrument

Due to a lack of questionnaires specifically designed to measure factors that impact providing bereavement support, The Australian College of Midwives developed the Perinatal Bereavement Care Confidence Scale (PBCCS) [25]. To understand the baseline bereavement knowledge, skills, self-awareness, and organizational support among staff working at our hospital, the study team adopted the validated PBCCS. Permission to utilize the questionnaire was requested and subsequently received on February 20th, 2024. The questionnaire required approximately 10 minutes to complete. Prior to participation, respondents provided informed consent, which outlined eligibility criteria and explained the study purpose and rationale.

   The 43-item questionnaire is conceptually grounded in the literature with a panel of experts supporting the content validity of the PBCCS; Bereavement care knowledge (15 items, 3 subscales) skills (9 items, 2 subscales), self-awareness (8 items, 2 subscales), and organizational support (11 items, 2 subscales). The internal consistency ranges from 0.753 to 0.871 [25]. Items are rated on a Likert-based scale of 1–5 with 1 representing Strongly Disagree and 5 representing Strongly Agree. As the questionnaire was originally developed for midwives and nurses in Ireland, some items were not relevant to the current U.S. healthcare system, some were repetitive while others were related to confidence/awareness on the needs of bereaved parents expecting their next baby. A total of 14 questions were thus removed for the current research (Supplemental 1). To assess staff perceptions on our hospital's practices, as well as barriers and facilitators in providing bereavement care, three items were added to the organizational support subscale (these are noted in Table 2).

   The current study questionnaire included a total of 32 items: Bereavement care knowledge (12 items, 3 subscales), Bereavement care skills (7 items, 2 subscales), Self-awareness (7 items, 2 subscales), and Organizational support (6 items, 2 subscale). The range of scores for bereavement care knowledge (12 items) is 12–60; a score of 12 representing the least level of support knowledge and 60 representing the most. Score range was calculated in a similar way for the other subscales. To obtain the total bereavement support confidence score, the mean score for each survey item was calculated as well as for each subscale (score range of 32–160). As the authors who developed the original survey [25] determined scoring 80% or higher was considered having sufficient level of confidence, a score of 128 or higher (achieving 80%) for this research was determined as sufficient confidence.

### Statistical analysis

All statistical analyses were performed in SAS version 9.4. Descriptive statistics included the demographics of survey responders and the survey responses. Data was presented in tables, including frequencies, mean and standard deviation for each survey item. Chi-square tests were performed to assess differences in demographic characteristics such as gender, age, highest level of education, years of working at our hospital and any formal education on perinatal bereavement support. Independent t-tests and one way analysis of variance (ANOVA) measured the mean difference in scores for two groups and multiple groups respectively. Incomplete item responses were treated as missing and excluded on a per-item basis; analyses were conducted using available data for each survey item. Missing responses were not imputed, and denominators therefore vary across each item. There were no predetermined effect sizes for a formal power calculation to determine the number of subjects required for statistical significance as a convenience sample was used. All analyses with resulting p-values <0.05 were considered statistically significant.

### Results

#### Participants

The survey was completed by 55 staff, out of which 46 had valid responses with a response rate of 83.64%. Most of the staff were female (n = 35, 89.74%) and belonged to age group of 30–39 years (43.59%) (Table 1). Most (48.72%) had high school or bachelor's degree followed by Master's or Doctorate degree (43.59%). More than half (51.28%) had been working

**Table 1. Respondent Demographics (n = 46).**

| Demographics | Category | Count (%) | p-value |
|---|---|---|---|
| Gender | Female | 35 (89.74%) | <.0001[a] |
| | Male | 4 (10.26%) | |
| | Missing | 7 | |
| Age (years) | 18-29 | 12 (30.77%) | 0.37 |
| | 30-39 | 17 (43.59%) | |
| | 40+ | 10 (25.64%) | |
| | Missing | 7 | |
| Highest Level of Education | High School/Bachelors | 19 (48.72%) | 0.003[a] |
| | Masters/Doctorate | 17 (43.59%) | |
| | Other | 3 (7.69%) | |
| | Missing | 7 | |
| Years Working at Our Hospital | 0-4 | 20 (51.28%) | 0.87 |
| | 5+ | 19 (48.72%) | |
| | Missing | 7 | |
| Years Working in Current Role | 0-4 | 15 (38.46%) | 0.15 |
| | 5+ | 24 (61.54%) | |
| | Missing | 7 | |
| Current Area of Practice | Emergency Department | 13 (33.33%) | 0.0004[a] |
| | Labor & Delivery | 19 (48.72%) | |
| | Maternity | 5 (12.82%) | |
| | Other | 2 (5.13%) | |
| | Missing | 7 | |
| Received Formal Education/Training on Perinatal Bereavement Support | Yes | 11 (28.21%) | 0.006[a] |
| | No | 28 (71.79%) | |
| | Missing | 7 | |
| Type of Education/Training | Formal Academic Education/Training | 9 (26.47%) | 0.0005[a] |
| | Informal Education/Training | 19 (55.88%) | |
| | None | 6 (17.65%) | |
| | Missing | 12 | |

[a]p < .05.

at our hospital for four or less years. Almost 50% of respondents worked in L&D, 33.33% were from ED and 12.82% from Maternity. More than 70% of respondents reported that they had not received any formal education on perinatal bereavement as compared to 28.21% of staff who reported receiving either informal or some bereavement education (p = 0.006).

Table 2 shows the mean score of each survey item within each PBCCS subscale. The lowest average score of 3.09 was found for the knowledge survey item "*I have been well prepared to provide perinatal bereavement support*". Similarly, the lowest average score (3.12) for self-awareness category was found for "*I am aware of my personal resources for bereavement support*". For bereavement support skills, the lowest score (2.86) was found for the survey item: "*I can easily respond to the needs of bereaved sibling when accompanying their parents*" and for organizational support, the lowest score (2.90) was for the item: "*I am aware of the passage of House Bill No. 6835 (Effective July 1, 2023) that requires our state's hospitals to give parents of babies who pass at birth at least 24 hours to be with their stillborn child.*"

Table 3 presents the total mean sum score for each PBCCS subscale along with the score range. Results showed low levels of bereavement support knowledge (46.45 ± 13.04), bereavement support skills (23.07 ± 7.31), self-awareness

**Table 2. PCBBS Survey Results by Subscale.**

| PBCCS Subscale | Survey Items | N | Mean | SD |
|---|---|---|---|---|
| Bereavement Support Knowledge | Perinatal loss is a traumatic event for bereaved parents | 46 | 4.67 | 1.01 |
| | Bereaved parents require the support of care team/ healthcare staff to cope with their loss | 45 | 4.56 | 1.03 |
| | I understand that grieving is a process | 46 | 4.59 | 1.07 |
| | I know how to provide the specific bereavement support needs of grieving mothers | 46 | 3.67 | 1.12 |
| | I understand the cultural needs of bereaved parents | 46 | 3.46 | 1.09 |
| | I understand the social needs of bereaved parents | 46 | 3.67 | 1.12 |
| | I do not know how to provide the specific bereavement support needs of grieving fathers | 46 | 3.15 | 1.11 |
| | I understand the religious needs of bereaved parents | 46 | 3.37 | 1.08 |
| | I know the referral system for additional bereavement support | 45 | 3.16 | 1.30 |
| | I have been well prepared to provide perinatal bereavement support | 45 | 3.09* | 1.31 |
| | There is a need for perinatal bereavement education for all professionals in healthcare | 46 | 4.65 | 0.74 |
| | All professionals at the hospital should receive perinatal bereavement continuing education. | 46 | 4.41 | 1.07 |
| Self-awareness | I am aware of the needs of recently bereaved parents | 41 | 3.39 | 1.16 |
| | I can easily empathize with grieving parents | 41 | 3.95 | 1.14 |
| | I am aware of my limitations in relation to the provision of perinatal bereavement support | 41 | 3.88 | 0.87 |
| | I am aware of my learning needs regarding bereavement support | 41 | 3.80 | 1.03 |
| | I am regularly engaged in reflective practice in relation to the provision of perinatal bereavement support | 41 | 3.15 | 1.11 |
| | I am aware of my personal resources for bereavement support | 41 | 3.12* | 1.23 |
| | Being aware of my need for support in relation to providing care for bereaved parents encourages me to seek help | 41 | 3.56 | 1.00 |
| Bereavement Support Skills | I have the skills to provide practical support to recently bereaved parents | 43 | 3.51 | 0.98 |
| | I do not have adequate perinatal bereavement support experience | 43 | 2.98 | 1.20 |
| | I have grief counselling skills for providing psychological support to bereaved parents | 43 | 2.88 | 1.00 |
| | I can comfortably listen to bereaved parents without trying to interrupt them. | 43 | 4.09 | 0.92 |
| | I can provide emotional care to bereaved parents | 43 | 3.74 | 1.00 |
| | I can provide spiritual care to bereaved parents | 43 | 3.00 | 1.09 |
| | I can easily respond to the needs of bereaved sibling when accompanying their parents | 43 | 2.86* | 1.10 |
| Organization Support | I have support from my workplace management in relation to providing bereavement support | 40 | 3.48 | 0.96 |
| | There is a clear policy in my ward/unit for the provision of bereavement support to parents | 40 | 3.30 | 0.97 |
| | Debriefing opportunities are always provided for me when required following a traumatic incident | 40 | 3.55 | 1.11 |
| | **I believe parents should have a suite/room dedicated for bereavement only. | 40 | 4.28 | 0.85 |
| | **I am aware of the passage of House Bill No. 6835 (Effective July 1, 2023) that requires our state's hospitals to give parents of babies who pass at birth at least 24 hours to be with their stillborn child. | 40 | 2.90* | 1.19 |
| | **Our hospital should provide CuddleCots, a flexible system to eliminate the need for cold rooms and allows families to spend precious time with their baby or child in a more comfortable setting. | 40 | 4.23 | 1.00 |

PBCCS = Perinatal Bereavement Care Confidence Scale. SD = Standard Deviation *The lowest mean score for each subscale **Items added to the original survey.

(24.85 ± 7.54) and organizational support (21.73 ± 6.08). The total mean sum score of 116 showed an inadequate level of confidence among the staff.

Scores for each PBCCS subscale between departments showed significant differences in mean score for self-awareness (p = 0.02). The mean score for ED staff was 3.17 while those working in L&D scored 3.83 for this subscale (Table 4).

No significant differences were found in other PBCCS subscales between the departments. Those reporting having a formal education were more likely to have greater self-awareness (p = 0.001) and bereavement support skills (4.23) as compared to those who did not (3.39) (Table 5).

**Table 3. PBCCS Subscale Results and Total Mean Scores.**

| PBCCS Subscale | Total Number of Items in Subscale | Score Range | 80% Score | Staff total sum scores Mean (SD) |
|---|---|---|---|---|
| Bereavement Support Knowledge | 12 | 12-60 | 48 | 46.45 (13.04) |
| Bereavement Support Skills | 7 | 7-35 | 28 | 23.07 (7.31) |
| Self-Awareness | 7 | 7-35 | 28 | 24.85 (7.54) |
| Organizational Support | 6 | 6-30 | 24 | 21.73 (6.08) |
| Total Score | 32 | 32-160 | 128 | 116.10 (33.97) |

SD = Standard Deviation, PBCCS = Perinatal Bereavement Care Confidence Scale. A score of 80% or higher for each subscale indicates "sufficient" subscale scores.

**Table 4. PBCCS Subscale Results by Department.**

| PBCCS Subscale | Department | | | p-value |
|---|---|---|---|---|
| | ED Mean (SD) | L&D Mean (SD) | Maternity Mean (SD) | |
| Bereavement Support Knowledge | 3.71 (0.86) | 3.98 (0.56) | 3.82 (0.98) | 0.71 |
| Bereavement Support Skills | 3.16 (0.61) | 3.44 (0.69) | 2.97 (0.31) | 0.31 |
| Self-Awareness | 3.17 (0.41) | 3.83 (0.34) | 3.71 (0.51) | 0.02[a] |
| Organizational Support | 3.32 (0.69) | 3.86 (0.56) | 3.80 (0.84) | 0.37 |

[a]$p<.05$, SD=Standard Deviation, PBCCS=Perinatal Bereavement Care Confidence Scale.

**Table 5. Sub-Analysis on Formal Bereavement Education.**

| PBCCS subscale | Formal education on perinatal bereavement support | | p-value |
|---|---|---|---|
| | Yes Mean (SD) | No Mean (SD) | |
| Bereavement Support Knowledge | 4.20 (0.68) | 3.74 (0.77) | 0.13 |
| Bereavement Support Skills | 3.74 (0.92) | 3.12 (0.53) | 0.15 |
| Self-Awareness | 4.23 (0.16) | 3.39 (0.43) | 0.001[a] |
| Organizational Support | 4.03 (0.59) | 3.55 (0.56) | 0.19 |

[a]$p<.05$, SD=Standard Deviation, PBCCS=Perinatal Bereavement Care Confidence Scale.

The survey also included the option for participants to submit comments or suggestions. A total of nine responses were received. One participant noted, *"Nurses have done an excellent job with the perinatal loss committee; more education and support is needed for residents and attendings."* Another emphasized the widespread training: *"Everyone that has any patient contact should be educated on this topic."* Two respondents specifically highlighted the need for training focused on grief, and another noting formal policies on training and workflow are limited. Another comment noted that the hospital has one Cuddle Cot, as well as a bereavement committee responsible for staff and patient education, hosting a memorial each year for patients and their families who experienced perinatal loss. They noted: *"this is so helpful to staff and patients in taking care of patients with perinatal loss."* The retrospective chart review identified 1,096 patients who had a visit or diagnosis code for stillbirth, miscarriage, neonatal/fetal demise, or an unplanned abortion from January 1st, 2021, to March 15th, 2024.

## Discussion

This quality improvement project needs assessment reported on the knowledge, attitudes, and processes that staff currently utilize in managing patients who experience perinatal loss at our community-based teaching hospital. The validated PBCCS [25] was utilized to survey staff employed in the ED, L&D, and Maternity Unit as frontline staff experiencing

perinatal loss and grieving parents. Our results indicated that a significant number of responders did not have formal education on perinatal bereavement support, highlighting a need for further education. Staff felt ill-equipped to provide perinatal bereavement support as a result. The survey results also highlighted that staff did not know how to respond to the needs of bereaved siblings and that they were not aware of personal resources available for bereavement support. Overall, from the assessment, the staff showed inadequate levels of confidence, with the highest scores for confidence in the L&D staff and lowest in the ED staff. Importantly, when staff did receive formal education on perinatal bereavement support, they experienced a significant increase in self-awareness when compared to those that did not.

Our findings underscore the critical need for structured education and resource availability to support frontline staff managing patients and families experiencing perinatal loss. The lack of formal education among a significant proportion of respondents likely contributes to the reported feelings of being ill-equipped to provide appropriate bereavement care. This is consistent with a recently published scoping review, which highlights that healthcare professionals with specialized training in perinatal bereavement exhibit greater confidence and competence in addressing the emotional and psychological needs of grieving families [26]. The disparity in confidence levels between L&D and ED staff further suggests the importance of tailoring educational interventions to the unique demands of each clinical setting.

An important aspect of perinatal loss management is addressing the needs of the entire family unit, including bereaved siblings. For families, coping with this loss highly depends on social and emotional support and this support can play a vital role in helping the families cope with grief and helps them navigate the healing process [27]. Haghighi et al. found individualized counseling directly after receiving the news can significantly aid in emotional recovery, underscoring the need for tailored psychological interventions [28]. The findings of Stanhope et al. further emphasize the need for external support systems in mitigating adverse postpartum health outcomes, highlighting how social and professional support influence recovery trajectories. Support can come from healthcare professionals, support groups, community organizations, family and friends [3]. Current staff knowledge and awareness in this area were notably insufficient, reflecting gaps identified in previous literature.

According to Limbo and Kobler [29], comprehensive family-centered care, which considers the emotional needs of siblings and extended family, is crucial for holistic bereavement support. However, our findings reveal that this critical component is often overlooked, reinforcing the necessity of nursing-targeted training programs to include not just the parents, but the whole family to ensure the provision of family-centered care. Importantly, our study identified that formal education not only improved staff confidence but also enhanced self-awareness, a key component in delivering empathetic care. These findings align with research by Johnson and Esplen et al., demonstrating the use of high-fidelity simulation, both reporting significant improvements in nursing and emergency staff knowledge, emotional resilience and the ability to establish therapeutic connections with patients [30,31].

To address these challenges, institutions should consider implementing evidence-based educational programs, such as perinatal bereavement workshops and simulation training, which have been shown to improve staff preparedness and patient outcomes [32]. Furthermore, ensuring access to institutional resources, including mental health services for staff and bereavement care pathways for families, could help bridge the identified gaps. Collaborative approaches that involve nursing, social workers, mental health professionals, and chaplaincy services may further enrich the support network for grieving families and staff alike. Most of the surveyors on the open comments agreed that information, formal training, an in service for bereavement, education and more staff member that can be part of the bereavement committees is the best way to promote confidence and increasing the comfort level in providing care.

Evidence-based programs augmenting the clinical environment for grieving parents have been reported by patients and hospital staff as crucial [11,14,15]. In addition to targeted hospital-wide education, leaders should prioritize the implementation of bereavement suites; located external to the maternity unit, this designated area provides a private, supportive care environment that offers resources and case management for those experiencing a loss. At our hospital, we are exploring the opportunity to create a bereavement suite to aid families via the provision of a supportive environment that is respectful, resource-driven, and family-centered.

Limitations of this study include assessing rates and perceptions at one community hospital and was limited to three departments. It is unclear whether the baseline knowledge of the current staff could have been impacted by study analysis and results due to the type of hospital, location, and size. Compared to the other hospitals in the area that are composed of many hospitals joined into one healthcare organization, these results could have varied if more staff or hospitals were included in this study. In addition, the validity and reliability scores were not re-evaluated for the modified scale, as the survey was originally developed to evaluate bereavement confidence, competencies, and skills among midwives in Ireland, rather than among acute care hospital staff in the U.S. who also care for patients experiencing perinatal loss. The intent of the current cross-sectional study was not to re-validate or evaluate the instrument for acute care hospital staff, but rather to use the existing tool pragmatically to assess staff confidence, competencies, and perceived skills within our specific clinical context.

This study is the first on the topic which will set the stage for additional research on understanding the existing state of bereavement care in hospitals in our state to assess the impact of such interventions on both provider confidence and patient outcomes. Additionally, exploring cultural and contextual variations in bereavement care practices could provide valuable insights for developing inclusive and effective support systems; especially due to the unpredictable maternal and child healthcare policy, program funding, and political environments we are currently experiencing [33].

## Conclusion

Perinatal loss is a profound and multifaceted life event, leaving a lasting impact on families and healthcare providers across the world. This global burden highlights both the challenges and the opportunities to strengthen bereavement care provision to better support the grieving and healing process for this at risk population. This quality improvement project highlights critical gaps in education, confidence, and resource awareness among frontline staff managing perinatal loss in a community-based teaching hospital. The findings underscore the pressing need for structured training programs and institutional resources to equip staff with the necessary skills to provide compassionate and effective bereavement support.

## Supporting information

**S1 File. Supplemental 1: 14 questions that were removed from PBCCS.**
(DOCX)

## Author contributions

**Conceptualization:** Carolina Garavito, Suzanne Rose, Josette Hartnett.

**Data curation:** Carolina Garavito, Shweta Karki, Suzanne Rose, Josette Hartnett.

**Formal analysis:** Shweta Karki, Suzanne Rose, Josette Hartnett.

**Investigation:** Carolina Garavito, Suzanne Rose, Josette Hartnett.

**Methodology:** Carolina Garavito, Shweta Karki, Suzanne Rose, Josette Hartnett.

**Project administration:** Suzanne Rose, Josette Hartnett.

**Resources:** Suzanne Rose, Josette Hartnett.

**Supervision:** Suzanne Rose, Josette Hartnett.

**Validation:** Shweta Karki, Suzanne Rose.

**Writing – original draft:** Carolina Garavito, Shweta Karki, Suzanne Rose, Josette Hartnett.

**Writing – review & editing:** Carolina Garavito, Shweta Karki, Suzanne Rose, Josette Hartnett.

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
