## [Decision Letter · Decision Letter 0]

6 Feb 2026

Dear Dr. Rose,

Thank you for submitting your manuscript to PLOS ONE. After careful consideration, we feel that it has merit but does not fully meet PLOS ONE’s publication criteria as it currently stands. Therefore, we invite you to submit a revised version of the manuscript that addresses the points raised during the review process.

We look forward to receiving your revised manuscript.

Kind regards,

JONATHAN BAYUO, PhD

Academic Editor

PLOS One

**Journal Requirements:**

1. Please ensure that your manuscript meets PLOS ONE's style requirements, including those for file naming. The PLOS ONE style templates can be found at https://journals.plos.org/plosone/s/file?id=wjVg/PLOSOne_formatting_sample_main_body.pdf and https://journals.plos.org/plosone/s/file?id=ba62/PLOSOne_formatting_sample_title_authors_affiliations.pdf  2. In the online submission form, you indicated that  “All de-identified data is publicly available at the following location: https://figshare.com/s/1e143cb02cde1e5f4ef4 and can also be provided with any written request to the corresponding author.” All PLOS journals now require all data underlying the findings described in their manuscript to be freely available to other researchers, either 1. In a public repository, 2. Within the manuscript itself, or 3. Uploaded as supplementary information.This policy applies to all data except where public deposition would breach compliance with the protocol approved by your research ethics board. If your data cannot be made publicly available for ethical or legal reasons (e.g., public availability would compromise patient privacy), please explain your reasons on resubmission and your exemption request will be escalated for approval.

**Additional Editor Comments:**

The authors present an interesting study regarding bereavement practices and staff competencies in managing perinatal loss. Kindly see the comments below to help strengthen the manuscript:

1. It is unclear why the authors label the study as a "pilot study". With the use of survey, a cross-sectional design will be more appropriate. Kindly consider this further.

2. Kindly add the absolute figures to the percentages mentioned in the abstract section.

3. The authors mention 11 questions were removed from the original instrument. It will be helpful, for methodological purposes, to list these questions in a table. Also, was the validity and reliability scores re-evaluated for the modified scale? This is extremely important.

4. The "omission of missing data" could use more details. Does the authors mean all responses that were incomplete were discarded?

5. Did the authors observe any variations across the participants from the L&D and ED? Such nuances, if observed, should be highlighted.

Reviewers' comments:

Reviewer's Responses to Questions

**Comments to the Author**

1. Is the manuscript technically sound, and do the data support the conclusions?

Reviewer #1: Yes

2. Has the statistical analysis been performed appropriately and rigorously?

Reviewer #1: Yes

3. Have the authors made all data underlying the findings in their manuscript fully available?

Reviewer #1: Yes

4. Is the manuscript presented in an intelligible fashion and written in standard English?

Reviewer #1: Yes

Reviewer #1: This is a well written manuscript with significant findings for improving perinatal loss management in hospital settings as well as highlighting the pressing need for structured training programs and institutional resources to equip staff with the necessary skills to provide compassionate and effective bereavement support.

I have only minor comments for clarification purposes.:

1. On page 4 line 109, could you provide the full meaning of the 'TEARDROP' program described as an interdisciplinary training on perinatal bereavement care, if there is.

2. On page 5 line 128 -129, could you clarify the statement 'This pilot study was conducted after receiving Institutional Review Board review and 'determination as exempt' what do you mean 'determination as exempt'?

3. On page 6 lines 162-164, 'As the questionnaire was originally developed for midwives and nurses in Ireland, some items (11 total) were not relevant to the current U.S. healthcare system and thus removed for the current research.' Could you clarify if these 11 items removed were within the same subscale? If so, as part of this tool development, does the developers allow for each subscale to be assessed as a separate component and thus removing these 11 items does impact on the total outcome score for the tool?

Thank you such important manuscript.

.

Reviewer #1: **Yes:**Mary Abboah-OffeiMary Abboah-OffeiMary Abboah-OffeiMary Abboah-Offei

---

## [Author Response · Author response to Decision Letter 1]

5 Mar 2026

Please see separate word document attached. Thank you

---

## [Editor Report · Decision Letter 1]

31 Mar 2026

Bereavement Practices and Staff Competencies on Perinatal Loss at a Community-Based Teaching Hospital

PONE-D-25-50589R1

Dear Dr. Rose,

We’re pleased to inform you that your manuscript has been judged scientifically suitable for publication and will be formally accepted for publication once it meets all outstanding technical requirements.

Kind regards,

JONATHAN BAYUO, PhD

Academic Editor

PLOS One

Additional Editor Comments (optional):

Thanks to the authors for addressing the comments raised.
---

## [Editor Report · Acceptance letter]

PONE-D-25-50589R1

PLOS One

Dear Dr. Rose,

I'm pleased to inform you that your manuscript has been deemed suitable for publication in PLOS One. Congratulations! Your manuscript is now being handed over to our production team.

Kind regards,

on behalf of

Dr. JONATHAN BAYUO

Academic Editor

PLOS One